# Neutrophil Adaptations upon Recruitment to the Lung: New Concepts and Implications for Homeostasis and Disease

**DOI:** 10.3390/ijms21030851

**Published:** 2020-01-28

**Authors:** Vincent D. Giacalone, Camilla Margaroli, Marcus A. Mall, Rabindra Tirouvanziam

**Affiliations:** 1Department of Pediatrics, Emory University School of Medicine, Atlanta, 30322 GA, USA; vincent.giacalone@emory.edu (V.D.G.); camilla.margaroli@gmail.com (C.M.); 2Center for CF & Airways Disease Research, Children’s Healthcare of Atlanta, Atlanta, 30322 GA, USA; 3Department of Pediatric Pulmonology, Immunology and Critical Care Medicine and Cystic Fibrosis Center, Charité-Universitätsmedizin Berlin, 13353 Berlin, Germany; marcus.mall@charite.de; 4Berlin Institute of Health (BIH), 10178 Berlin, Germany

**Keywords:** margination, metabolism, scavenging, stress response, transcription

## Abstract

Neutrophils have a prominent role in all human immune responses against any type of pathogen or stimulus. The lungs are a major neutrophil reservoir and neutrophilic inflammation is a primary response to both infectious and non-infectious challenges. While neutrophils are well known for their essential role in clearance of bacteria, they are also equipped with specific mechanisms to counter viruses and fungi. When these defense mechanisms become aberrantly activated in the absence of infection, this commonly results in debilitating chronic lung inflammation. Clearance of bacteria by phagocytosis is the hallmark role of neutrophils and has been studied extensively. New studies on neutrophil biology have revealed that this leukocyte subset is highly adaptable and fulfills diverse roles. Of special interest is how these adaptations can impact the outcome of an immune response in the lungs due to their potent capacity for clearing infection and causing damage to host tissue. The adaptability of neutrophils and their propensity to influence the outcome of immune responses implicates them as a much-needed target of future immunomodulatory therapies. This review highlights the recent advances elucidating the mechanisms of neutrophilic inflammation, with a focus on the lung environment due to the immense and growing public health burden of chronic lung diseases such as cystic fibrosis (CF) and chronic obstructive pulmonary disease (COPD), and acute lung inflammatory diseases such as transfusion-related acute lung injury (TRALI).

## 1. Introduction

Neutrophils comprise the largest proportion of circulating leukocytes in the human body and maintain a major presence in organs such as the lung. Consequently, despite being considered as terminally differentiated and endowed with a short lifespan after leaving the bone marrow, they are a major player in innate immunity. Their hallmark function is clearance of debris and pathogens through phagocytosis but they exhibit a diverse array of other immune functions. In addition to the direct phagocytosis of bacteria [1] and fungi [2], they limit the spread of microbes by releasing neutrophil extracellular traps (NETs) made of DNA through a process known as NETosis [3]. Although neutrophils are professional killers, they also have significant capacity to modulate the function of other immune cells. For example, through secretion of arginase-1 (Arg1) they suppress T-cell proliferation in the airways of cystic fibrosis (CF) patients [4] and limit T-cell function in the tumor microenvironment [5]. Similarly, the release of neutrophil elastase (NE) has been attributed to the alteration of macrophage function by the cleavage of Toll-like receptors (TLRs) and cytokines [6], T-cell function through the cleavage of surface co-receptors [7], and the modulation of secreted antibodies [8]. Diverse effector functions of secreted proteins, some of which are summarized in Table 1, are key for neutrophil adaptability and their far-reaching effects on immune responses.

Not only do neutrophils exhibit greater functional diversity than once thought [9], but they demonstrate the ability to reprogram and adapt to local microenvironments upon recruitment to tissues. This is in contrast with the conventional view holding these cells as terminally differentiated once released from the bone marrow. For example, a recent report details neutrophil reprogramming in a model of atherosclerosis via oxidized calmodulin-dependent protein kinase II driving a pro-inflammatory phenotype alongside the suppression of homeostatic transcription factors [10]. Neutrophil adaptation in the context of CF lung disease has been reviewed extensively [11,12] and has incited further investigation into neutrophil reprogramming in other diseases. Of particular interest is neutrophil adaption upon recruitment to the lung, due to the role this organ plays as a major neutrophil reservoir [13,14]. There is a rapidly increasing prevalence of chronic inflammatory lung diseases such as asthma and chronic obstructive pulmonary disease (COPD) in the global population, in part due to increased exposure to air pollution [15]. In addition, aging populations are faced with increased risk for common nosocomial infections such as bacterial pneumonia, which results in sustained neutrophil recruitment to the lung but reduced efficacy in clearing infections [16]. Neutrophilic inflammation is also a key component of progressive lung damage in patients with CF [17,18], which is one of the most common fatal hereditary diseases [19]. A greater understanding of pathological features of neutrophils in such lung pathologies is critical to improving treatment options for both acute and chronic inflammatory diseases.

## 2. Homeostasis

Like every tissue in the body, the lung is characterized by a specific immune profile. Circulating neutrophils are retained in the lung microvasculature, forming a reservoir defined as the lung-marginated neutrophil pool. The first observation of leukocyte sequestration in the lung microvasculature was reported in 1910 by F.W. Andrewes during a series of lectures held for the Royal College of Physicians in London [31]. Subsequently, the use of radiolabeling and adoptive transfer techniques allowed the identification of a marginated neutrophil pool in the capillary bed of the lungs, which was found to be in a dynamic equilibrium with the circulating pool [13,32,33,34,35]. The causes for neutrophil margination in the lung capillary bed had been initially attributed to several factors, including the size of the capillary vessel, the time required for a neutrophil to adapt its shape, hemodynamics, and to physical interactions with the lung endothelium [36,37,38,39,40]. However, differences in blood flow in the pulmonary arterioles and venules and the waterfall effect support a major role for physical interactions with the endothelial layer rather than a merely hemodynamic cause [41,42]. Indeed, the potential roles of selectins and integrins in orchestrating neutrophil margination in the lung capillary bed have been investigated. However, one should consider that the endothelium is a highly dynamic tissue programmed to respond quickly to internal and external cues. Therefore, its surface protein landscape changes upon different stimuli. At steady state, CD11/CD18 or L-selectins may not play a central role in the retention of neutrophils into the lung capillary bed [43,44]. However, other selectins cannot be excluded since treatment with fucoidin, an inhibitor of selectin-mediated adhesion, has been shown to partially disrupt neutrophil margination in the lung microvasculature [45]. Recent advances in intravital microscopy and genetically engineered mouse models have improved our understanding of neutrophil biology related to their margination to the capillary bed. Interestingly, a study by Devi et al. implicated the CXCR4/CXCL12 axis in neutrophil margination in the lung [23], pointing at CXCR4 as a retention marker (rather than a marker for aging neutrophils, as it is sometimes viewed).

Since the first observation of leukocyte retention made by Andrewes in 1910 [31], a physiological role for neutrophil margination to the lung has been sought. Several groups have hypothesized that neutrophil margination acts as a protective mechanism to de-prime and sequester activated neutrophils, thus preventing further damage [46,47,48]. Recently, supported by the observation that marginated lung neutrophils express the major histocompatibility complex II and interact with B cells in the lung microvasculature [49], Granton et al. hypothesized that the lung—as with the spleen and liver—may act as an immunological niche [50]. While both theories may support physiological roles to maintain homeostasis and control the immune response, more studies will be needed to elucidate the role for neutrophil margination in the lung microvasculature at steady-state and upon stress responses (Figure 1). The large surface area of the lung environment is indeed constantly exposed to inhaled pathogens and other environmental stimuli, and having abundant neutrophils in the immediate vicinity allows for an efficient and rapid innate immune response.

## 3. Stress Response

Unlike some of the longer-lived leukocytes featured in both arms of the immune system, such as mast cells in the innate system and memory T cells and long-lived plasma cells in the adaptive system, neutrophils do not maintain long-term tissue residence. However, they are invaluable for their ability to respond rapidly and in massive manner to almost any type of stress inflicted upon peripheral tissues. The hallmark function of neutrophils is microbial clearance, especially of bacteria, by phagocytosis. It is now well understood how bacteria are captured and digested internally [1,51], but neutrophils are also well equipped to clear viruses [52,53] and fungi [2], as well as contribute to defense against parasites [54].

### 3.1. Bacterial Infections

Responding to bacterial infections is the most well-characterized function of neutrophils. With the increased focus on innate immunity in recent years, we are now learning more about the complexity of pathogen identification and clearance in specialized areas such as the lung. Although neutrophils are characterized by pre-programmed functions, their ability to carry out these functions is highly dependent on the specific conditions of their microenvironment. For example, a recent report by Lei et al. details the differences in neutrophil response to Group A *Streptococcus* in the lungs compared to the skin. While this bacterial species is efficiently cleared from the lungs of mice by a nicotinamide adenine dinucleotide phosphate oxidase (NOX)-dependent mechanism, clearance is impaired in the skin [55]. This discrepancy may be due to creation of an anoxic environment in the skin, which favors the growth of this organism but hampers oxidative burst by neutrophils. Indeed, NOX-dependent generation of reactive oxygen species (ROS) is a crucial component of a neutrophilic response [1]. The CD200 receptor has been shown to play a role in driving lung pathology during influenza infection, as blocking this receptor attenuated macrophage-associated inflammation [56]. However, blocking this receptor on neutrophils during pulmonary *Francisella tularensis* infection in mice augmented infection by reducing ROS production [21]. Although there are scenarios in which it would be beneficial to counteract excessive ROS production, for bacterial and fungal infections [57] as well as viral infections [58], this finding demonstrates the potential benefit of boosting ROS production in certain cases. For example, boosting innate immune responses following influenza infection in mice by overexpression of granulocyte-macrophage colony-stimulating factor in the lungs was found to protect against *Staphylococcus aureus*-induced pneumonia by enhancing ROS production in alveolar macrophages but not neutrophils. Neutrophils were, however, essential for protection and the mice did not experience excessive inflammation resulting from elevated ROS production [59]. In another example, ROS production was enhanced by treatment with an angiotensin-converting enzyme inhibitor, which promoted the killing of methicillin-resistant *S. aureus* [60]. This mechanism may not only apply to enhancing the antimicrobial response. For example, it has also been implicated in the stimulation of wound healing through enhancing the differentiation of pro-resolution macrophages in the liver [61].

Of equal importance to ROS in the destruction of bacteria are reactive nitrogen species such as nitric oxide (NO) [62] which are produced by neutrophils to a high degree in diseased airways [63]. Inducible nitric oxide synthase (iNOS) is the enzyme complex responsible for generating NO using arginine as a substrate [25] and has long been known to be highly activated in neutrophils in response to bacterial infection [64]. However, neutrophils are not the sole source of NO produced in tissues, as it is also produced by endothelial cells [65] and macrophages. NO production by all cells can be inhibited by Arg1, which competes with iNOS for arginine as a substrate [66], and is actively secreted by neutrophils in chronic diseases such as CF [4] and cancer [67]. While RNS are important microbicidal mediators, they can have detrimental effects when released from activated neutrophils and other cells. In a study by Kumar et al. looking at septic patients with confirmed bacterial infections, neutrophils were found to have increased iNOS activity. Nitrite, a metabolite of NO, was measured in the plasma and found to inversely correlate with lung function [68]. While lung function was likely impacted by other aspects in this severe pathological condition, these findings emphasize the potency of a neutrophilic response in impacting the function of organs such as the lungs. In an in vitro model of sepsis, Shelton et al. found that neutrophil iNOS activity contributed to leakage across an endothelial barrier with evidence that peroxynitrite, produced by NO reacting with O_2_^-^, mediates this effect [69]. While this model did not directly use bacterial challenge, it employed mixtures of cytokines important for sepsis in humans, which is defined as “life-threatening organ dysfunction caused by a dysregulated host response to infection” [70], in which neutrophilic inflammation is an important component [71].

Another antimicrobial mechanism involves the release of histone-bound DNA complexed with primary granule proteins, such as NE and myeloperoxidase (MPO), in the form of NETs [72]. Formation of NETs is regulated by a complex pathway requiring histone citrullination by peptidyl arginine deiminase 4 (PAD4) followed by decondensation of the chromatin [73], which has more recently been shown to be promoted by histone acetylation using broadly acting inhibitors of histone deacetylase [74]. Mechanisms independent of PAD4 have also been described [75,76]. NETosis has been classically viewed as a cell death pathway but neutrophils have been shown to maintain viability and anti-bacterial functionality following NET release [77,78]. Although the exact role of NETosis in cellular fate is still under debate [79,80], this process has demonstrated importance in the neutrophilic response to bacterial lung infections, including bacterial pneumonia. In a study of patients with ventilator-associated pneumonia, the presence of NETs was assessed by measuring complexes of DNA and MPO. NET presence was found to be elevated in acute respiratory distress syndrome (ARDS) patients with ventilator-associated pneumonia compared to ARDS alone, and correlated with both bacterial burden and CXCL8 [81]. Considering that MPO has been found to be elevated in the airways of ARDS patients [82] and that this enzyme is associated with lung damage in CF [83,84], NET-associated MPO may be a contributing factor in progression of disease in ARDS. Studies of NETosis are also contributing to a better understanding of neutrophil plasticity. Although typically thought of as having minimal transcriptional activity, NETosis has been shown to be dependent on transcription. Khan and Palaniyar demonstrated that inhibition of transcription using Actinomycin D attenuated NETosis in response to the bacterial stimulants lipopolysaccharide and ionomycin [85]. This finding implicates transcription as a potential target of inhibition for treating NET-related pathologies. Another therapeutic target related to NETs formation is Type I interferon. A recent finding suggests that Type I interferon-driven NETosis may promote respiratory *P. aeruginosa* infections in mice by providing a scaffold to support biofilm production [86]. Interruption of interferon signaling may, therefore, be useful for suppressing infection by biofilm-capable organisms, but care would need to be taken not to increase susceptibility to viral infection. Finally, when considering the cost-benefit of NETosis, with regards to its anti-microbial properties and its association with disease pathology, one should consider the balance between the number of neutrophils present at the site of inflammation vs. those undergoing NETosis. While this quantification may be challenging to measure, especially in vivo, it may clarify its impact on the fitness of the host in a specific inflammatory setting.

### 3.2. Viral Infections

Although neutrophils are most thoroughly studied in the context of antibacterial responses, a growing body of literature demonstrates their importance in responding to viral infection, as well. The adaptations they undergo in this role, especially when recruited to the airways, have major implications for disease outcome and eventual resolution of inflammation since neutrophil-driven innate immune mechanisms can mount a rapid antiviral response even in the absence of memory B- and T-cell responses [87,88]. The fine balance of neutrophilic inflammation in response to respiratory tract viral infections is exemplified by a variety of findings that may appear contradictory. For example, a recent study demonstrated that activation of the NOD-, LRR- and pyrin domain-containing protein 3 (NLRP3) inflammasome improves survival during influenza A virus (IAV) infection in mice by recruiting neutrophils via interleukin-1β [89]. However, neutrophils do not provide a universal protective effect during influenza infection. Studies using aged mice have demonstrated overall higher neutrophil presence in the lung during IAV infection, but impaired migration towards the chemoattractant CXCL1 and reduced expression of the corresponding receptor CXCR2 on bone marrow neutrophils. Depleting neutrophils after infection was also shown to promote survival without impairing clearance [22]. Since overall neutrophil recruitment was not shown to be impacted while migration towards specific chemoattractant gradients was dysregulated, this demonstrates the need for more targeted immunomodulatory therapies focused on neutrophilic responses. A new potential target described in a recent study using a murine model of IAV infection is BCL6. This transcriptional regulator was shown to suppress neutrophil apoptosis specifically in airway neutrophils near the site of infection, while cells in the bone marrow and in circulation were unaffected. Mice with a myeloid cell deficiency in BCL6 exhibited improved survival and reduced inflammation when infected with IAV [20]. Modulating neutrophilic inflammation may therefore provide clinical benefit when chronic lung disease results from influenza infection [90]. However, careful consideration will likely be needed to determine which patients would benefit from an enhanced neutrophil response during viral infections, and which would benefit more from suppressing neutrophil activity.

While neutrophil adaptations in response to influenza infection are some of the best-studied, important findings with other viruses are continuing to direct much-needed focus toward the innate immune response to viral infections (Figure 2). It was recently demonstrated in a mouse model of respiratory syncytial virus (RSV) infection that while signaling through myeloid differentiation primary response 88 (MyD88) and TIR-domain-containing adapter-inducing interferon-β (TRIF) is essential for neutrophil recruitment to the lung, signaling through mitochondrial antiviral-signaling protein (MAVS) is required for neutrophil activation and the production of key mediators including matrix metalloproteinase 9 (MMP-9), MPO, and NE [91]. RSV infection is associated with severe neutrophilic inflammation, sometimes contributing to mortality [92]. Resolving neutrophilic inflammation is therefore as important as suppressing the infection, and a recent report suggests that leukocyte-associated Ig-like receptor 1 (LAIR-1) fills this role during RSV infection. This receptor is not expressed on circulating neutrophils, but may be upregulated upon migration into tissues and subsequent activation [26]. RSV-infected mice lacking functional LAIR-1 exhibited greater neutrophil influx into the airways but had no indication of enhanced viral clearance [27].

In addition to bacterial infection, NETosis has been described in the context of neutrophil responses to viral infection in previous reviews [93,94]. Studies of NETosis in response to respiratory viruses further establish the important implications of neutrophils in responding to viral infections, for both enhancing clearance and contributing to pathology. Murano et al. demonstrated that RSV is capable of inducing NETosis by the classical PAD4-mediated pathway. They also observed possible virion trapping demonstrated by co-localization of extracellular DNA lattices and primary granule proteins, including NE and MPO, with RSV F protein [95]. NETs have also demonstrated efficacy in the neutrophil response against HIV. Saitoh et al. demonstrated that the initiation of NETosis by signaling through TLR-7 and TLR-8 promotes trapping and inactivation of HIV through activity of the effector proteins MPO and α-defensin [96]. However, viral-induced NETosis can also have a detrimental impact on the host. In a mouse model of influenza infection, neutrophils from the bronchoalveolar lavage fluid were highly NETotic when co-incubated with infected epithelial cells and contributed substantially to lung injury [97]. In addition, blood neutrophils from influenza-infected patients have demonstrated a high propensity for NETosis when stimulated ex vivo, and NETosis was found to increase vascular permeability using an in vitro model [98]. Moreover, extracellular host DNA released in accordance with rhinovirus infection has been observed to correlate with a shift towards a type 2 immune response in humans, typically associated with allergic disease. Using a murine model of rhinovirus infection, Toussaint et al. then demonstrated that infection promotes NET formation and inhibition of NETosis reduced type 2 immune pathology [99]. While sometimes effective in limiting viral infection, further research is needed to determine if the risks of anti-viral NETosis outweigh the potential reward as a therapeutic target.

### 3.3. Fungal Infections

In addition to bacteria and viruses, neutrophils can mount a powerful response to fungi as well. One of the major culprits behind respiratory fungal infections is *Aspergillus fumigatus*, which promotes potent neutrophilic inflammation [100] induced in part by the regulation of the Von Willebrand factor via cleavage by a disintegrin and metalloproteinase with a thrombospondin type 1 motif, member 13 [101]. In addition to their role in clearance of spores and hyphae [102], neutrophils play a key role in regulating the antifungal adaptive immune response. In neutropenic mice infected with *A. fumigatus*, dendritic cells were found to accumulate in the lungs, but had impaired homing to the mediastinal lymph nodes. Dendritic cells were also found to lack expression of surface costimulatory molecules, but this defect was rescued by coincubation with neutrophils following *Aspergillus* exposure [103]. Considering that neutrophils are essential for guarding the lungs against fungal pathogens it is surprising that CF patients, who experience widespread neutrophilic inflammation in the airways, are highly susceptible to respiratory fungal infections [104]. The cause may be neutrophils themselves. Neutrophils acquire a dysfunctional phenotype in the CF airways which hampers their ability to clear bacteria, despite attaining a high state of activation where they exocytose their primary granules [105]. Exocytosis of primary granules releases NE into the extracellular environment, and this protease has been found to cleave the pattern recognition receptors dectin-1 and 2 [24], which are important phagocytic receptors for fungal pathogen-associated molecular patterns such as β-glucan [106,107]. Cleavage of these receptors by NE inhibited the antifungal response in infected mice [24], and the airway fluid from CF patients has been found to have a high prevalence of extracellular NE [108]. Neutrophil elastase drives multiple aspects of CF lung disease, including increased mucus production and impairment of mucociliary clearance [109]. Both outcomes promote colonization of the airways by opportunistic fungal pathogens such as *Aspergillus*, which has a complex array of interactions with airway mucins [110]. Considering that this protease is being actively secreted by neutrophils undergoing pathological conditioning in the CF airways [105], with similar dysfunctions observed in other respiratory diseases [111], NE inhibition may offer a therapeutic option for treating pulmonary fungal infections. Indeed, pyrimidine derivatives in some NE inhibitors have antifungal properties [112].

## 4. Neutrophils in Chronic Respiratory Pathologies

We are now building a greater understanding of how neutrophil responses to infectious challenge, especially in the lung, are far more complex than simply locating and clearing microbes. Their responsiveness to non-infectious challenge, for example with allergen and smoke exposure, is also a potent factor in innate immune responses. The ability of neutrophils to quickly resolve these challenges or contribute to pathology has major implications for both organ-specific and systemic health.

In the absence of infection, neutrophilic inflammation can be initiated and result in severe inflammatory pathologies. When neutrophils are recruited to the lungs in the absence of infection, in genetic disorders such as CF [12], or in diseases linked to environmental conditions such as COPD [113], they can cause extensive damage through release of their destructive granule contents such as NE and MPO [12,113]. They also exhibit potent immunomodulatory capabilities whereby they can substantially alter the immune balance of various environments [4,114]. In addition to ongoing tissue damage and altered adaptive immune responses, chronic neutrophilic pathologies are also characterized by neutrophil dysfunction where these cells are not able to effectively conduct their normal duties of debris and microbe clearance. A better understanding of how neutrophilic pathologies are initiated and how they might be corrected is essential for treating patients with rare diseases like CF that currently have few anti-inflammatory treatment options, as well as widespread diseases like asthma and COPD which are becoming ever-larger public health burdens each year.

### 4.1. Cystic Fibrosis (CF)

CF is a severe monogenic multiorgan disease that affects multiple epithelial organs, with the majority of morbidity and mortality due to airway disease [115]. Neutrophils have a central role in the development and persistence of airway disease and their role in pathology has been studied extensively [11,12]. Importantly, they have been shown to develop a unique inflammatory phenotype after recruitment into the CF airway lumen, where they maintain viability and exocytose their primary granules but have reduced ability to phagocytose bacteria in a distinct fate now called GRIM (granule releasing, immunomodulatory, and metabolically active) neutrophils [105,116,117]. Exocytosis of the primary granules, which are usually sequestered in the cytoplasm, results in release of effectors including NE, Cathepsin G, MPO and Arg1, which has been found to correlate with disease progression in mice with CF-like lung disease and young children and older patients with CF [118,119,120]. This pro-inflammatory phenotype with reduced ability to clear pathogens poses an intriguing paradox and is becoming a focal point in addressing CF lung disease. This failure of a major defense mechanism in the lungs is likely a contributing factor to the high susceptibility of CF patients to common environmental bacteria [121]. Directly causing lung damage through protease and oxidase release further implicates neutrophil dysfunction in being a major problem in CF lung disease that warrants additional research [109]. Prior studies of metabolic reprogramming in CF airway neutrophils may offer some explanation to how this dysfunctional phenotype is acquired. CF airway neutrophils activate the mechanistic target of rapamycin (mTOR) pathway [122] and increase expression of the Glut1 glucose transporter [123], which is controlled by mTOR [124]. Activation of the mTOR pathway and expression of Glut1 promotes utilization of glucose in CF airways [125]. Another aspect of reprogramming in CF airway neutrophils is increased production of the regulatory protein resistin [126], which is closely tied to insulin resistance [127]. While resistance to insulin impairs the uptake of glucose by cells, anabolic reprogramming of neutrophils in the CF airways enables them to effectively take in and utilize glucose to fuel pro-survival pathways [105,122]. The downside of this adaptation is that resistin decreases the ability of neutrophils to kill bacteria by inhibiting ROS production and actin polymerization, as noted for CF-associated pathogens *P. aeruginosa* and *S. aureus* [128].

### 4.2. Asthma

Asthma is among the most common chronic diseases in children and adults, and typically viewed as a Th2-mediated allergic disease featuring profound eosinophilic inflammation [129]. However, there is a growing focus on neutrophilic inflammation in non-atopic asthma [130,131]. One of the contributing factors to neutrophilic asthma is respiratory infections. For example, Patel et al. demonstrated that young asthmatics who tested positive for *Chlamydia pneumoniae* exhibited elevated neutrophil counts and CXCL8, a powerful neutrophil chemoattractant, in bronchoalveolar lavage fluid [132]. Following on this discovery, Patel and Webley then used a mouse model of respiratory *Chlamydia* infection to demonstrate that airway neutrophils produce large amounts of the inflammatory mediators hepoxilin A_3_ and histamine [133]. However, the propensity of neutrophils to exacerbate inflammatory conditions in asthma is neither restricted to bacterial pathogens nor to the airways. Neutrophils isolated from the peripheral blood of asthmatics were found to have enhanced secretion of CXCL8 in response to the viral surrogate and TLR agonist R848 compared to non-asthmatics [134], which may provide an explanation for why patients with viral respiratory tract infections are more likely to experience treatment failures [135]. While most therapies are directed at treating eosinophil-mediated allergic asthma, options for developing neutrophil-directed asthma treatments have been investigated. The macrolide antibiotic clarithromycin has previously shown promise in suppression of neutrophilic inflammation in patients with refractory asthma as shown by a reduction in sputum CXCL8 concentration and neutrophil presence [136]. While a variety of therapies targeting neutrophilic asthma have proven ineffective, as reviewed recently by Seys et al. [137], an increased understanding of neutrophil dysregulation in diseased airways will hopefully enable much-needed therapeutic breakthroughs.

### 4.3. Chronic Obstructive Pulmonary Disease (COPD)

COPD has emerged as one of the most common causes of morbidity and mortality worldwide. Caused mostly by exposure to environmental factors such as tobacco smoke and pollution [138], COPD also depends on underlying genetic predispositions [139]. As with CF, lung disease in COPD is characterized by a heavy neutrophil component [140], with extracellular NE being associated with severity of disease and exacerbations [141,142,143]. In addition, it was observed by Chrysanthopoulou that exposure to cigarette smoke induces NET formation which contributes to lung fibrosis [144]. Recently, Genschmer et al. highlighted a new mechanism of NE-induced lung damage, showing that NE in the airways of COPD patients is localized on the surface of exosomes [145] forming an active NE pool resistant to inhibition by α1-antitrypsin. While this study provides strong evidence for the damage phenotype, Garratt et al. showed that NE inhibition by α1-antitrypsin mitigated epithelial repair [146], suggesting a potential physiological role for NE-associated exosomes in mediating epithelial repair. Indeed, NE acts on epithelial surfaces by triggering pro-reparative epidermal growth factor signaling [147,148].

Exposure to tobacco smoke and pollution has been well studied, but a new trend showing intriguing effects on innate immunity is the use of electronic cigarettes. Exposure to vaporized nicotine and e-liquid base from electronic cigarettes was found several years ago to impair bacterial killing by neutrophils [149], as with a more recent finding using neutrophils exposed to total particulate matter from conventional cigarette smoke [150]. While neutrophil bactericidal capacity was found to be impaired, the activation and production of inflammatory markers and mediators were augmented upon exposure to e-cigarette vapor extract, as demonstrated by Higham et al. Neutrophils showed increased expression of surface CD11b and CD66b and secreted more MMP-9 and CXCL8, while NE activity was found to be increased in culture conditions [151]. In addition, blood neutrophils isolated from e-cigarette users showed elevated susceptibility to the induction of NETosis, which was supported by an increased presence of NET-related proteins in the sputum of these subjects compared to non-smokers [152]. These findings of neutrophil activation in response to e-cigarette components closely mirror what is observed in COPD due to smoke and air pollution exposure and provide solid evidence for similarly detrimental impact on lung health due to induction of neutrophilic inflammation.

### 4.4. Transfusion-Related Acute Lung Injury (TRALI)

Transfusion-related acute lung injury (TRALI) is one of the leading causes of transfusion-induced morbidity and mortality. It manifests within 6 hours following transfusion with fever and hypoxemia, followed by bilateral pulmonary edema [153,154]. Observed since the 1950s but only defined as a distinct syndrome in 1983 [155], TRALI is thought to be triggered by two factors—known as the “two-hit theory” [156]—implicating the status of the transfused material and the vulnerability of the recipient. Studies by Toy et al. [157] and Popovsky [158] identified several risk factors associated with TRALI, including mechanical ventilation, sepsis and female-derived blood. The role of neutrophils in TRALI is highlighted by their increased recruitment in the pulmonary capillary bed and into the alveolar space [159], the presence of neutrophil-derived proteases, and the production of reactive oxygen species, which have been described as mediators of lung injury [160,161,162]. Although the precise temporal sequence of the events of the “two-hit theory” remains unclear, it is thought that neutrophils primed by the patient’s underlying condition adhere to the pulmonary endothelium and that platelets play a major role in further activating them [163]. Several components have been identified as possible mediators of the second hit. Antibodies against HLA class I or class II, and human neutrophil antigens in donor and recipient blood have been implicated in neutrophil activation and subsequent capillary leakage into the alveoli, resulting in pulmonary edema [164,165,166,167]. Alterations following increased storage time of red blood cells prior to transfusion have been investigated as a possible cause of neutrophil activation in pre-clinical models [168] and in clinical studies [169,170]. Differences detected in pre-clinical models did not occur in patients, suggesting that longer storage—within the clinically allowed time—is not sufficient to induce the same type of neutrophil activation in humans as in animal models. Bioactive lipids, formed during storage, have also been investigated for their ability to promote TRALI, yielding contradicting results [168,171], further suggesting a role for factors related to donor specificity [172,173,174]. The complement system [175], gut microbiota [176], and other soluble factors, such as osteopontin [177], have also been suggested to play a role in neutrophil accumulation and capillary leakage leading to TRALI. Of note, certain clinical subsets of TRALI do not implicate neutrophils as the key pathogenic cell subset as illustrated by the occurrence of TRALI in neutropenic patients [178,179] and in neutrophil-depleted animal models [180]. In conclusion, the mechanisms underlying the recruitment, activation, and pathological role of neutrophils need to be better understood.

## 5. Conclusions

Recent advances in understanding neutrophil biology in health and disease emphasize the plasticity of these cells. Despite comprising the largest proportion of circulating leukocytes, studying the molecular mechanisms of neutrophil function is still a relatively new endeavor in the field of immunology. The identification and clearance of microbes by neutrophils is now well understood, but this represents only a small sample of their functional capabilities. These functional capabilities depend on an array of signaling and effector proteins (Table 1).

There is now a growing interest in the contribution of neutrophils in chronic disease, especially those relating to the lungs. First, while neutrophilic inflammation has long been known to have a role in CF and COPD, new mechanisms are uncovered by which they may contribute to lung damage. Second, new studies of asthma are shedding light on pathological mechanisms that are driven by a neutrophilic response despite typically being thought of as a type 2-dominated eosinophilic airway disease. Third, considering the monogenic disease CF where the focus has historically been on the mutated CF Transmembrane conductance Regulator ion channel (expressed primarily in epithelial cells), recent studies have indicated that early and sustained neutrophil recruitment to the airways and activation in the mucostatic environment of the CF lung is a major factor in the initiation and progression of lung disease [18,181,182]. With our improved understanding of neutrophil contribution to chronic diseases and new data demonstrating metabolic and transcriptional adaptions of neutrophils in these circumstances, neutrophil-directed therapies may soon become an option for the innovative treatment of diseases characterized by chronic neutrophilic inflammation, such as CF and COPD, or acute inflammation such as TRALI.

## Figures and Tables

**Figure 1 ijms-21-00851-f001:**
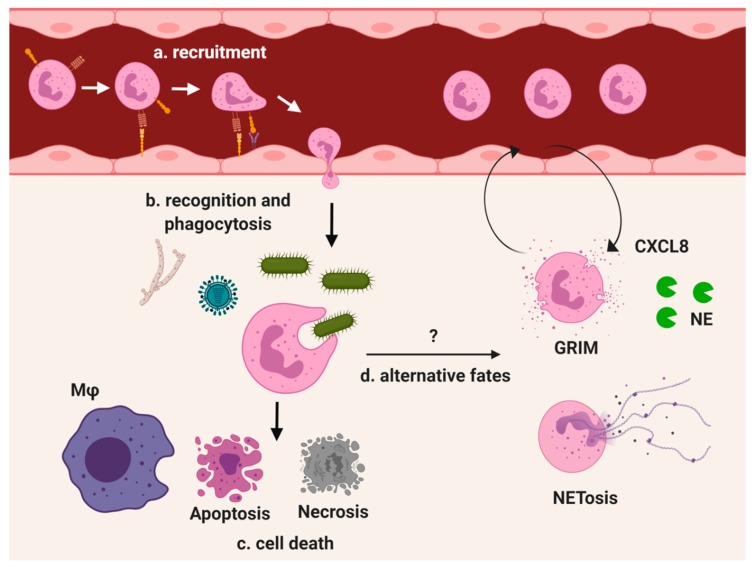
Overview of neutrophil recruitment and response in the lung. As a major neutrophil reservoir, the lungs are an important environment in the study of neutrophil biology, both at homeostasis and in responding to inflammatory stimuli. (**a**) Endothelial cells upregulate P-selectin to bind P-selectin glycoprotein ligand-1 on circulating naive neutrophils. Upon slowing down, neutrophil β2 integrin binds with higher affinity to ICAM-1 on the endothelial cell surface followed by extravasation into the tissue. (**b**) Recruited neutrophils recognize pathogen-associated molecular patterns from all types of pathogens by surface pattern recognition receptors. Phagocytosed pathogens are degraded internally by fusion of the granules with the phagosome. (**c**) Neutrophils quickly apoptose and are cleared by tissue macrophages. (**d**) Neutrophil recruitment can also lead to alternate fates. Dysregulated neutrophil responses in diseases such as CF and COPD include the GRIM (granule releasing, immunomodulatory, and metabolically active) phenotype which exhibits active degranulation but impaired pathogen clearance. The release of NE damages host tissue while sustained CXCL8 production drives further neutrophil recruitment. Neutrophils can also expel their DNA through NETosis, but may survive and retain phagocytic capability.

**Figure 2 ijms-21-00851-f002:**
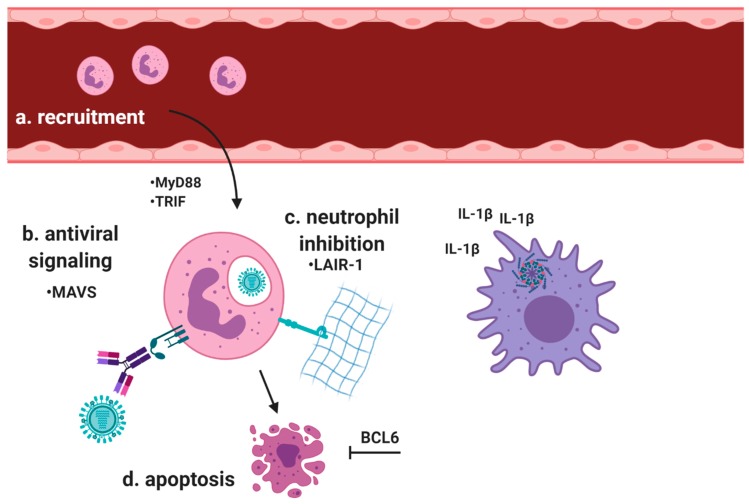
Neutrophil responses to viral infection. Neutrophils have an important role in antiviral immunity. (**a**) Neutrophils are recruited to sites of viral infection in the lung via signaling through MyD88 and TRIF. Interleukin-1β production by the NLRP3 inflammasome in resident antigen presenting cells drives recruitment. (**b**) once in the tissue, MAVS signaling initiates neutrophil activation and production of inflammatory mediators. Neutrophils engulf antibody-bound virions via surface Fc receptors. (**c**) The inhibitory receptor LAIR-1 binds collagen and suppresses neutrophil activity. (**d**) Transcriptional regulators such as BCL6 suppress apoptosis and represent a potential target for enhancing neutrophil-mediated antiviral immunity.

**Table 1 ijms-21-00851-t001:** Diverse roles of neutrophil signaling and effector proteins. Neutrophil effector proteins, such as proteases and phagocytic receptors, and receptors involved in chemotaxis, contribute to the plasticity of neutrophils in driving inflammation, promoting resolution, or maintaining homeostasis. Abbreviations: Arg1, arginase-1; iNOS, inducible nitric oxide synthase; LAIR-1, leukocyte-associated Ig-like receptor 1; MPO, myeloperoxidase; MMP-9, matrix metalloproteinase-9; NOX, nicotinamide adenine dinucleotide phosphate oxidase; NE, neutrophil elastase; oxCAMKII, oxidized calmodulin-dependent protein kinase II.

Protein	Role	Function
Arg1	pro/anti-inflammatory	suppresses T-cell proliferation [4]
BCL6	pro/anti-inflammatory	suppresses neutrophil apoptosis [20]
CD200R	anti-inflammatory	attenuates oxidant production by neutrophils [21]
CXCR2	pro-inflammatory	promotes chemotaxis as receptor for CXCL1 [22]
CXCR4	homeostatic	promotes retention in bone marrow/lung as receptor to CXCL12, [23]
Dectin-1	pro-inflammatory	promotes phagocytosis of fungi [24]
iNOS	pro-inflammatory	supports the generation of nitric oxide [25]
LAIR-1	anti-inflammatory	suppresses neutrophil recruitment [26,27]
MPO	pro-inflammatory	supports generation of hypochlorous acid [28]
MMP-9	pro-inflammatory	degrades the extracellular matrix [29]
NOX	pro-inflammatory	supports the generation of superoxide [1]
NE	pro-inflammatory	degrades phagocytosed microbes [1] and extracellular matrix [30]
oxCAMKII	pro-inflammatory	activates STAT1 and generation of inflammatory mediators [10]

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
