# Peer review of "Neutrophil Adaptations upon Recruitment to the Lung: New Concepts and Implications for Homeostasis and Disease"

_ijms, 2020, doi:10.3390/ijms21030851_

Round 1

Reviewer 1 Report

In the current manuscript, the authors review and discuss the role of neutrophils in the immune response to infection and in pathological conditions. A special attention is paid to neutrophil adaptation upon recruitment from blood circulation in the specific context of infections and diseases affecting the lungs.

This manuscript is well written and is presented in a clear, original and sound manner. The authors have undertaken a thorough and up-to-date revision of the published literature. As such, it importantly leads the reader to: (1) carefully reposition neutrophils as central cells in immunity and disease, and (2) acknowledge these cells as potential targets for therapeutic approaches. 

In my opinion, the current manuscript will still benefit from additional improvements that I will present subsequently:

1) In the legend of Figure 1, the authors refer for the first time to GRIM neutrophils, without elucidating what they are. The authors only elucidate this further on the text in page 16. And still here, they do not state what the acronym stands for.

I would advise the authors to correct these omissions. As a suggestion, the authors could already in the legend of figure 1 produce a small and brief explanation to what GRIM neutrophils are.

2) In my opinion, figure 1 is not of great help for the full comprehension of the manuscript. I do not think it is a figure of easy and immediate visual comprehension. It absolutely requires its legend to be understood in its plenitude. For instance, it is not clear by the visual inspection of the figure what should be perceived as a normal neutrophil function in the immune response, or in opposition, a dysregulated action of this cellular type. In addition, the figure and its legend may be as well misleading: for instance, NETosis is only here presented in the context of a dysregulated pathological neutrophil function. In my opinion, this figure could be improved for instance, by separating more distinctively what may be the normal and physiological actions of neutrophils in immune response/inflammation, from what may be functionally dysregulated functions in other immune/pathological conditions that may modulate/subvert neutrophil function.

3) In page 9, arginase 1 is presented as a competitive inhibitor of iNOS. Is this the correct biochemical form of presenting the current knowledge? What I can perceive from the literature is that both enzymes compete for the same substrate, arginine. But it strikes me as odd to read that they are competitive inhibitors of each other.

4) In page 11, the authors should consider adding references for the second sentence of the section of “Viral infections”. More precisely, references should account for: “since neutrophil-driven innate immune mechanisms can mount a rapid antiviral response even in the absence of memory B and T cell response”.

5) In page 12, the authors present Bcl6 as a promoter of PMN apoptosis (at least in the context of the influenza virus infection), and this idea is consistently reiterated in the legend of Figure 2 and in Table 1. For stating this, the authors acknowledge their reference #79 (Zhu et al (2019)). Still, the reading of this reference leads one to take the opposite conclusion, i.e., that Bcl6 is a transcriptional repressor of apoptotic genes and that as such, Bcl6 inhibits neutrophil apoptosis. If my interpretation of the data is correct, the information presented on Bcl6 should be amended throughout the manuscript.

6) My final consideration goes to table 1. In this manuscript, it is presented to concisely recapitulate data on the several factors and receptors herein presented as critical for neutrophil actions. Still, in my opinion this should alternatively be presented earlier in the introduction of the manuscript to introduce the factors and receptors that will be critically mentioned in the manuscript. In addition, and regardless of its position in the paper, the table should as well include the most relevant references that account for the information presented in the table.

Author Response

REVIEWER 1

In the current manuscript, the authors review and discuss the role of neutrophils in the immune response to infection and in pathological conditions. A special attention is paid to neutrophil adaptation upon recruitment from blood circulation in the specific context of infections and diseases affecting the lungs.

This manuscript is well written and is presented in a clear, original and sound manner. The authors have undertaken a thorough and up-to-date revision of the published literature. As such, it importantly leads the reader to: (1) carefully reposition neutrophils as central cells in immunity and disease, and (2) acknowledge these cells as potential targets for therapeutic approaches.

In my opinion, the current manuscript will still benefit from additional improvements that I will present subsequently.

We appreciate the reviewer’s’ feedback and suggestions. We carefully considered all critiques and amended the manuscript and illustrations accordingly.

1) In the legend of Figure 1, the authors refer for the first time to GRIM neutrophils, without elucidating what they are. The authors only elucidate this further on the text in page 16. And still here, they do not state what the acronym stands for.

I would advise the authors to correct these omissions. As a suggestion, the authors could already in the legend of figure 1 produce a small and brief explanation to what GRIM neutrophils are.

We thank the reviewer for this comment and made the relevant corrections to the legend of Figure 1 [pages 7-8].

2) In my opinion, figure 1 is not of great help for the full comprehension of the manuscript. I do not think it is a figure of easy and immediate visual comprehension. It absolutely requires its legend to be understood in its plenitude. For instance, it is not clear by the visual inspection of the figure what should be perceived as a normal neutrophil function in the immune response, or in opposition, a dysregulated action of this cellular type. In addition, the figure and its legend may be as well misleading: for instance, NETosis is only here presented in the context of a dysregulated pathological neutrophil function. In my opinion, this figure could be improved for instance, by separating more distinctively what may be the normal and physiological actions of neutrophils in immune response/inflammation, from what may be functionally dysregulated functions in other immune/pathological conditions that may modulate/subvert neutrophil function.

We took into account these critiques and attempted to clarify typical vs. alternative fates of neutrophils in the modified version of Figure 1 [pages 7-8].

3) In page 9, arginase 1 is presented as a competitive inhibitor of iNOS. Is this the correct biochemical form of presenting the current knowledge? What I can perceive from the literature is that both enzymes compete for the same substrate, arginine. But it strikes me as odd to read that they are competitive inhibitors of each other.

We thank the reviewer for this comment and made the relevant correction to the text [page 10].

4) In page 11, the authors should consider adding references for the second sentence of the section of “Viral infections”. More precisely, references should account for: “since neutrophil-driven innate immune mechanisms can mount a rapid antiviral response even in the absence of memory B and T cell response”.

We thank the reviewer for this comment and added 2 references to support this statement [page 12].

5) In page 12, the authors present Bcl6 as a promoter of PMN apoptosis (at least in the context of the influenza virus infection), and this idea is consistently reiterated in the legend of Figure 2 and in Table 1. For stating this, the authors acknowledge their reference #79 (Zhu et al (2019)). Still, the reading of this reference leads one to take the opposite conclusion, i.e., that Bcl6 is a transcriptional repressor of apoptotic genes and that as such, Bcl6 inhibits neutrophil apoptosis. If my interpretation of the data is correct, the information presented on Bcl6 should be amended throughout the manuscript.

We apologize for this misstatement and thank the reviewer for setting this record straight. The sentence has been amended to reflect this [page 13].

6) My final consideration goes to table 1. In this manuscript, it is presented to concisely recapitulate data on the several factors and receptors herein presented as critical for neutrophil actions. Still, in my opinion this should alternatively be presented earlier in the introduction of the manuscript to introduce the factors and receptors that will be critically mentioned in the manuscript. In addition, and regardless of its position in the paper, the table should as well include the most relevant references that account for the information presented in the table.

We placed Table 1 in the beginning of the manuscript, as suggested. It is now referred to in the Introduction [pages 3-4].

Reviewer 2 Report

This is a well written review article highlighting advances in neutrophil biology in the lung with focus on pertinent lung diseases including cystic fibrosis and COPD.

Author Response

REVIEWER 2

This is a well written review article highlighting advances in neutrophil biology in the lung with focus on pertinent lung diseases including cystic fibrosis and COPD.

We thank the reviewer for this positive comment and believe the additional edits in this revised version will improve readability and coverage of this review.

Reviewer 3 Report

The article addresses a major problem in the field of neutrophil function.

Please discuss about the NETs and viral infection.

Page 10, line 8: At the end of this sentence: “Another antimicrobial mechanism involves ..... a process called NETosis”, please add reference entitled: “Neutrophil extracellular traps kill bacteria”, by Brinkmann V et al. 2004,

Page 19 line 13 It was reported for the first time by the Chrisanthopoulou et al., that exposure to cigarette smoke, induces NET formation.

The paper about the role of neutrophil in lung diseases in not fully without the section about the transfusion associated lung injury (TRALI), in which neutrophil play a critical role.

Author Response

REVIEWER 3

The article addresses a major problem in the field of neutrophil function.

We thank the reviewer for this positive comment.

Please discuss about the NETs and viral infection.

We thank the reviewer for this comment and added a section related to NETs and viral infection, as suggested [page 14].

Page 10, line 8: At the end of this sentence: “Another antimicrobial mechanism involves ..... a process called NETosis”, please add reference entitled: “Neutrophil extracellular traps kill bacteria”, by Brinkmann V et al. 2004,

We thank the reviewer for this comment and added the reference, as suggested [page 11].

Page 19 line 13 It was reported for the first time by the Chrisanthopoulou et al., that exposure to cigarette smoke, induces NET formation.

We thank the reviewer for this comment and made mention of this study, as suggested [page 21].

The paper about the role of neutrophil in lung diseases in not fully without the section about the transfusion associated lung injury (TRALI), in which neutrophil play a critical role.

We thank the reviewer for this comment and added a section on TRALI, as suggested [pages 22-23].

Reviewer 4 Report

This is a good review of the recent advances of the participation of neutrophils in the mechanisms inflammation focusing on the lung, primarily in
chronic lung diseases such as cystic fibrosis and chronic obstructive  pulmonary disease. 

Author Response

REVIEWER 4

This is a good review of the recent advances of the participation of neutrophils in the mechanisms inflammation focusing on the lung, primarily in chronic lung diseases such as cystic fibrosis and chronic obstructive pulmonary disease.

We thank the reviewer for this positive comment and believe the additional edits in this revised version will improve readability and coverage of this review.